# Bi-frequency operation in a membrane external-cavity surface-emitting laser

**Jake Daykin**[1]\*, Jonathan R. C. Woods[2], Roman Bek[3], Michael Jetter[4], Peter Michler[4], Ben Mills[5], Peter Horak[5], James S. Wilkinson[5], Vasilis Apostolopoulos[1]

**1** School of Physics and Astronomy, University of Southampton, Southampton, Hampshire, United Kingdom, **2** Aquark Technologies, Romsey, Hampshire, United Kingdom, **3** Twenty-One Semiconductors GmbH, Neckartenzlingen, Germany, **4** Institute for Semiconductor Optics and Functional Interfaces, University of Stuttgart, Stuttgart, Germany, **5** Optoelectronics Research Centre, University of Southampton, Southampton, Hampshire, United Kingdom

\* J.Daykin@soton.ac.uk

## Abstract

We report on the achievement of continuous wave bi-frequency operation in a membrane external-cavity surface-emitting laser (MECSEL), which is optically pumped with up to 4 W of 808 nm pump light. The presence of spatially specific loss of the intra-cavity high reflectivity mirror allows loss to be controlled on certain transverse cavity modes. The regions of spatially specific loss are defined through the removal of Bragg layers from the surface of the cavity high reflectivity mirror in the form of crosshair patterns with undamaged central regions, which are created using a laser ablation system incorporating a digital micromirror device (DMD). By aligning the laser cavity mode with the geometric centre of the loss patterns, the laser simultaneously operated on two Hermite-Gaussian spatial modes: the fundamental $HG_{00}$ and the higher order $HG_{11}$ mode. We demonstrate bi-frequency operation over a range of pump powers and sizes of spatial loss features, with a wavelength separation of approximately 5 nm centred at 1005 nm.

## 1 Introduction

Coherent and tunable continuous wave (CW) laser sources that operate simultaneously on two discrete optical frequencies see potential applications in a variety of fields, such as spectroscopy [1], communications [2, 3], metrology [4, 5], and THz generation [6–8].

Generating bi-frequency operation in semiconductor quantum well lasers has been an area of active research in recent years [9–14]. Previously bi-frequency operation in single cavity Vertical External-Cavity Surface-Emitting Lasers (VECSELs) has been demonstrated through the use of techniques such as: the inclusion of an intra-cavity birefringent filter [15, 16], the deposition of a loss inducing mask onto, or through the machining of, the VECSEL gain structure [7, 8]. In [14] we utilise a spatial loss inducing, laser ablated region on the surface of a high reflectivity cavity mirror to force bi-frequency operation. Analysis and optimisation of noise in bi-frequency VECSELs has been performed in [8, 12, 15, 17]. Bi-frequency VECSELs have also been used as coherent and tunable THz sources through the utilisation of the beat note between the two modes [7, 8].

**Data Availability Statement:** All data supporting this study is openly available from the University of Southampton repository in Ref 37. doi: 10.5258/SOTON/D2628.

**Funding:** VA would like to thank the Engineering and Physical Sciences Research Council for the grant EP/T001046/1 titled "UK National Quantum Technology Hub in Sensing and Timing". https://www.ukri.org/councils/epsrc/ The funders had no role in study design, data collection and analysis, decision to publish, or preparation of the manuscript.

Membrane External-Cavity Surface-Emitting Lasers (MECSELs) are optically pumped semiconductor quantum well lasers based on VECSEL technology, but lacking the intra-cavity distributed Bragg reflector (DBR), that were first demonstrated by Yang et al. in 2015 [18]. MECSELs also share many advantages present in VECSELs, such as narrow emission line-width due to its class A dynamics and low Schawlow-Townes limit [9, 19–21] and potential to generate high quality, diffraction limited beams with $M^2 < 1.05$ [22, 23]. The main advantages of MECSELs are: better heat extraction and faster fabrication, both of these advantages stem from the fact that MECSELs are much thinner structures than VECSELs.

Here we combine a cavity geometry needed for bi-frequency operation with a MECSEL gain chip. We utilise spatial loss inducing, laser ablated masks on the surface of a high reflectivity cavity mirror. The laser ablation machining of the mirror was performed using a Digital Micromirror Device (DMD) [14]. The spatially dependent loss present in the MECSEL cavity allows for the loss of particular transverse cavity modes to be controlled, and thus promotes the oscillation of two specific spatial modes. Additionally, due to the fact that both modes share the same external cavity and because there is an overlap between the modes in the gain region, the noise in the system is common to both modes and coherence can be enforced between them [7, 15].

Using a MECSEL instead of a VECSEL gain chip gives us the following notable advantages: (a) MECSELs can be fabricated usually 10 times faster than VECSELs, due to the lack of an intra-cavity DBR. (b) The whole emission bandwidth of the active region can be used, again, due to the lack of the DBR. (c) Less complexity and greater flexibility in material combinations due to no longer needing to lattice-match the active region to the DBR, leading to the potential for new wavelengths. (d) The lack of an intra-cavity DBR in MECSELs allows for improved thermal management, and thus higher efficiencies and output powers [24–26]. These advantages give promise for future rapid prototyping of bi-frequency sources with higher output powers and with wider wavelength selection and tunability [27].

## 2 Experimental methods

### 2.1 MECSEL gain sample

The approximately 1 μm thick gain structure is fabricated using Metallo-Organic Chemical Vapour Deposition (MOCVD) and is comprised of 10 strain compensated InGaAs/GaAsP quantum wells, GaAs buffer layers to position the wells at the nodes of the field, and GaInP capping layers to prevent oxidisation. The gain structure is sandwiched between two 350 μm thick SiC heat spreaders and maintained at 16˚C using a peltier controlled by a thermo-electric cooler (TEC) module, which removes heat via the water-cooled copper heat-sink in which the gain sample is mounted. The MECSEL gain structure used throughout this work is of the same growth and layer list as that presented in [24, 28], which has previously been shown to lase laterally as a membrane quantum well laser (MQWL).

### 2.2 Ablated mirror masks

The masked mirror used in this work has previously been utilised in a VECSEL cavity to demonstrate bi-frequency operation; a full description of the ablation process can be found in Woods et al. 2019 [14]. Our loss inducing ablated mirror masks are produced using an experimental set-up that has previously been explored in detail in the works of Xie, Mills and Heath [29–33]. The set-up utilises 1 mJ, 150 fs pulses from a Ti:Sapphire laser which are spatially homogenised and then directed onto a digital micro-mirror device (DMD) (Texas Instruments DLP3000), which shapes the beam before being focused through a microscope objective onto the surface of the mirror, as shown schematically in Fig 1B.

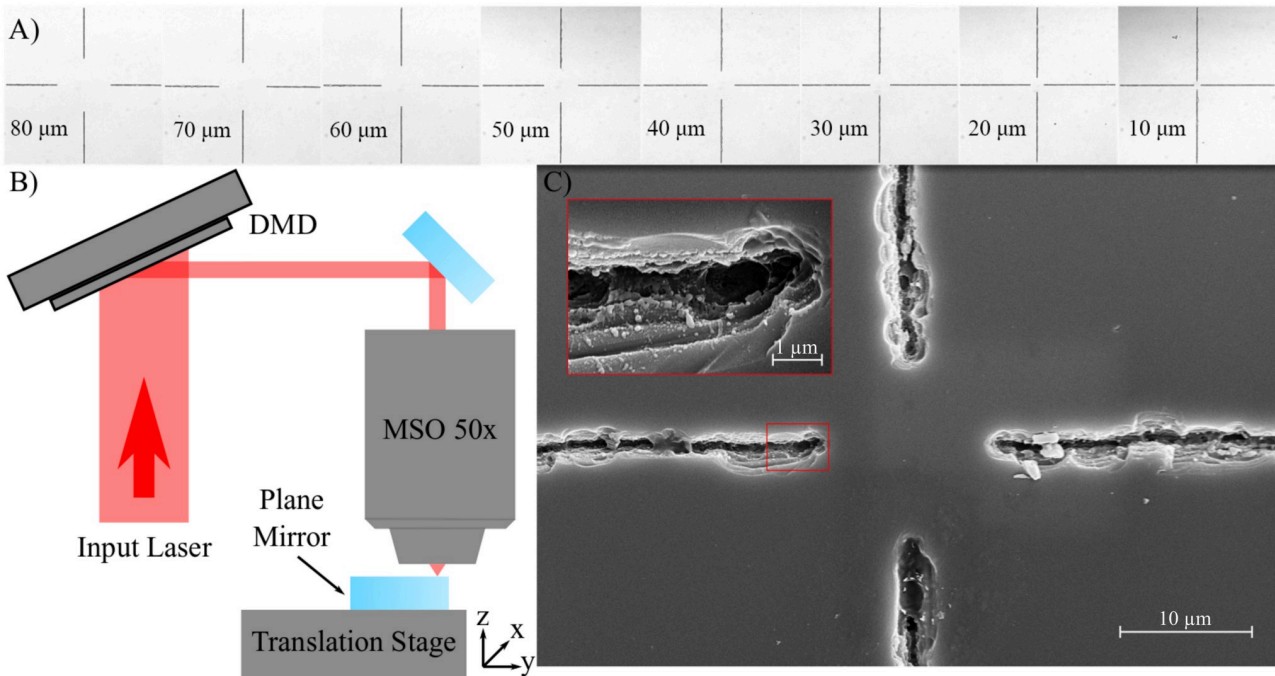

**Fig 1. Figure reprinted from J. Woods, et al. 2019 [14]** Optical (A) and electron (C) microscope images of the ablated crosshair patterns with widths ranging from 10 to 80 μm and (B) a schematic diagram of the DMD-enabled laser ablation setup which utilises a pulsed input laser and a 50× microscope objective to ablate the surface of the mirror.

The mirror masks used in this work are all crosshair shaped with unablated central regions. We utilise five different masks where the diameter of the unablated central region ranges from 10 to 50 μm, as shown in Fig 1A. A scanning electron microscope (SEM) image of an ablated mask can be seen in Fig 1C. While the ablation is not uniform, it has been found, through the use of a laser-based reflectivity set-up similar to that shown in [34], that the ablated areas have a minimum reflectivity of 6.2% ± 0.1%, and therefore introduce greater than 90% loss into the incident cavity modes in areas overlapping with the mask.

## 2.3 Cavity design

The MECSEL external cavity and experimental characterisation setup are shown schematically in Fig 2. The cavity is formed around the MECSEL gain structure in a linear cavity configuration with length 74 mm, consisting of a 12.8 mm diameter, 75 mm radius of curvature concave output coupler mirror at one end with reflectivity R = 99.0%, and a 6 mm diameter planar high reflectivity (HR) mirror (R > 99.9%), onto which the loss inducing masks are machined, at the other. The optics used in our MECSEL cavity are non-standard; rather than using a pair of curved mirrors to force a beam waist at the gain sample, the use of a planar HR mirror causes the waist instead to be on the masked HR mirror itself. This has the benefit of reducing the total area that requires ablation and also removes the need to design masks to account for mode astigmatism, but has the disadvantage of removing the gain sample from the waist of the cavity and increases the pump power necessary to reach lasing threshold due to a larger cavity spot size and therefore less gain per unit area.

The gain structure is optically pumped by a multimode fiber coupled diode laser, which operates at a wavelength of 805.4 nm. The pump laser is directed onto the gain structure from

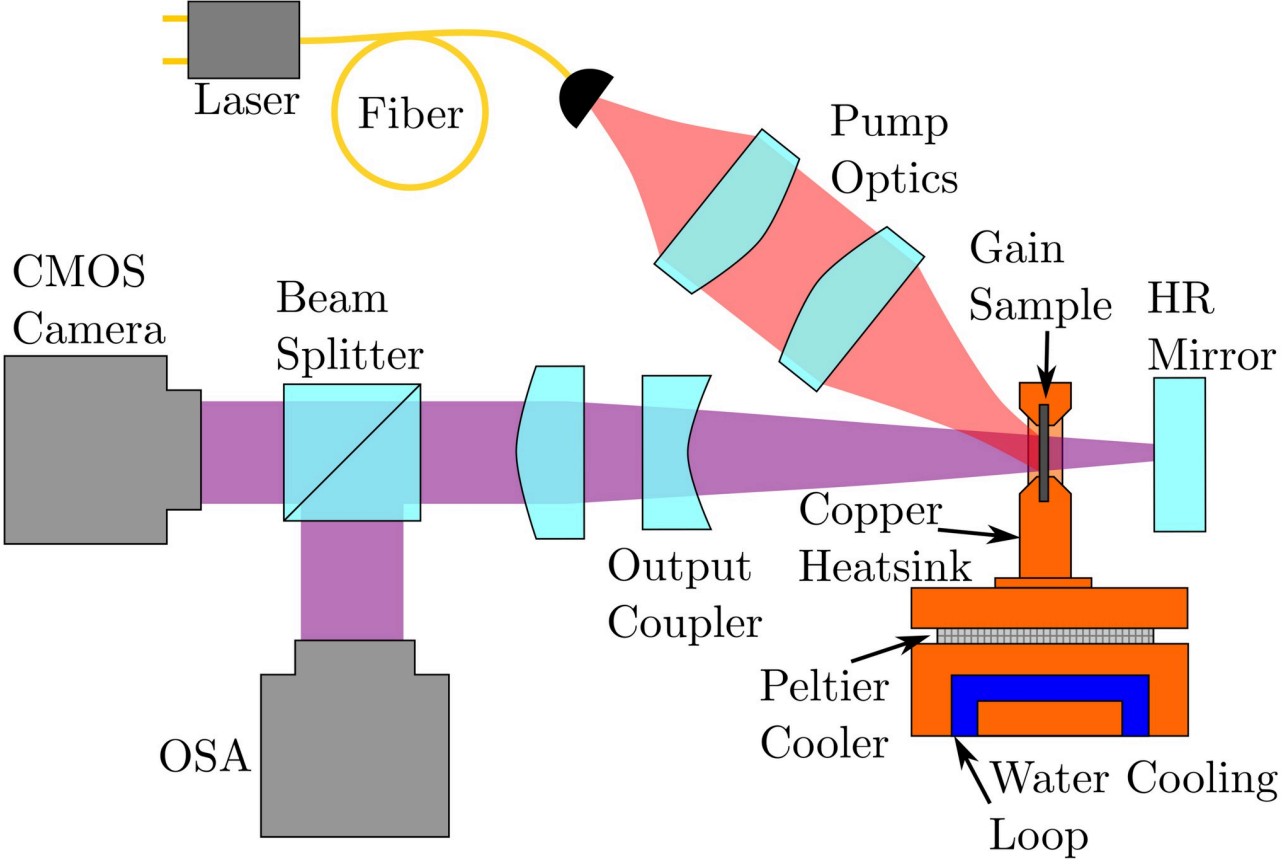

**Fig 2. Schematic of the bi-frequency MECSEL cavity.** The pump laser is capable of delivering 35 W through a 200 μm diameter fiber. The pump optics consist of a 20 mm collimation lens and a 40 mm focusing lens, generating a pump spot of radius 184 μm. The ablated masks are located on the high reflectivity (HR) mirror. OSA = Optical Spectrum Analyser.

a 200 μm core diameter step-index fiber using a 20 mm focal length collimation lens and a 40 mm focusing lens, leading to a pump spot with radius of 184 μm. The radius, $R_{nm}$, of a Hermite-Gaussian (HG) mode, $HG_{nm}$, is a function of the radius of the $HG_{00}$ mode, $R_{00}$, and is given by:

$$R_{nm} = R_{00} \times \sqrt{1 + n + m} \tag{1}$$

where n and m are integers that represent the mode indices, therefore the radius of the $HG_{11}$ mode is found to be $\sqrt{3}$ larger than that of the $HG_{00}$. The ablated masks are designed such that most cavity modes will overlap with the ablated regions and therefore experience high enough loss that prevents lasing; the modes which are still able to propagate within the cavity, other than the fundamental $HG_{00}$, are the $HG_{11}$, $HG_{33}$, $HG_{55}$, etc. It is expected that due to greater diffraction losses on the higher order modes, and through the design of the cavity, only the $HG_{11}$ mode will be able operate along side the $HG_{00}$ mode.

The $HG_{00}$ and $HG_{11}$ modes are calculated to have beam radii on the gain structure of 67 μm and 116 μm respectively; while on the masked HR mirror, the modes have radii of 52 μm and 90 μm respectively. The overlap of the pump spot and cavity modes must be sufficiently large to support the generation of both modes, and thus the gain structure is over-pumped.

## 3 Results

### 3.1 Loss estimations

Fig 3A shows the simulated overlap between the $HG_{00}$ and $HG_{11}$ modes with the loss features of the mask while the cavity spot is centred on an ablated mirror mask of each different mask width. In order to estimate the losses imposed on the cavity modes by the ablated mirror masks, we perform overlap integrals between the modes with each of the five masks. Using the

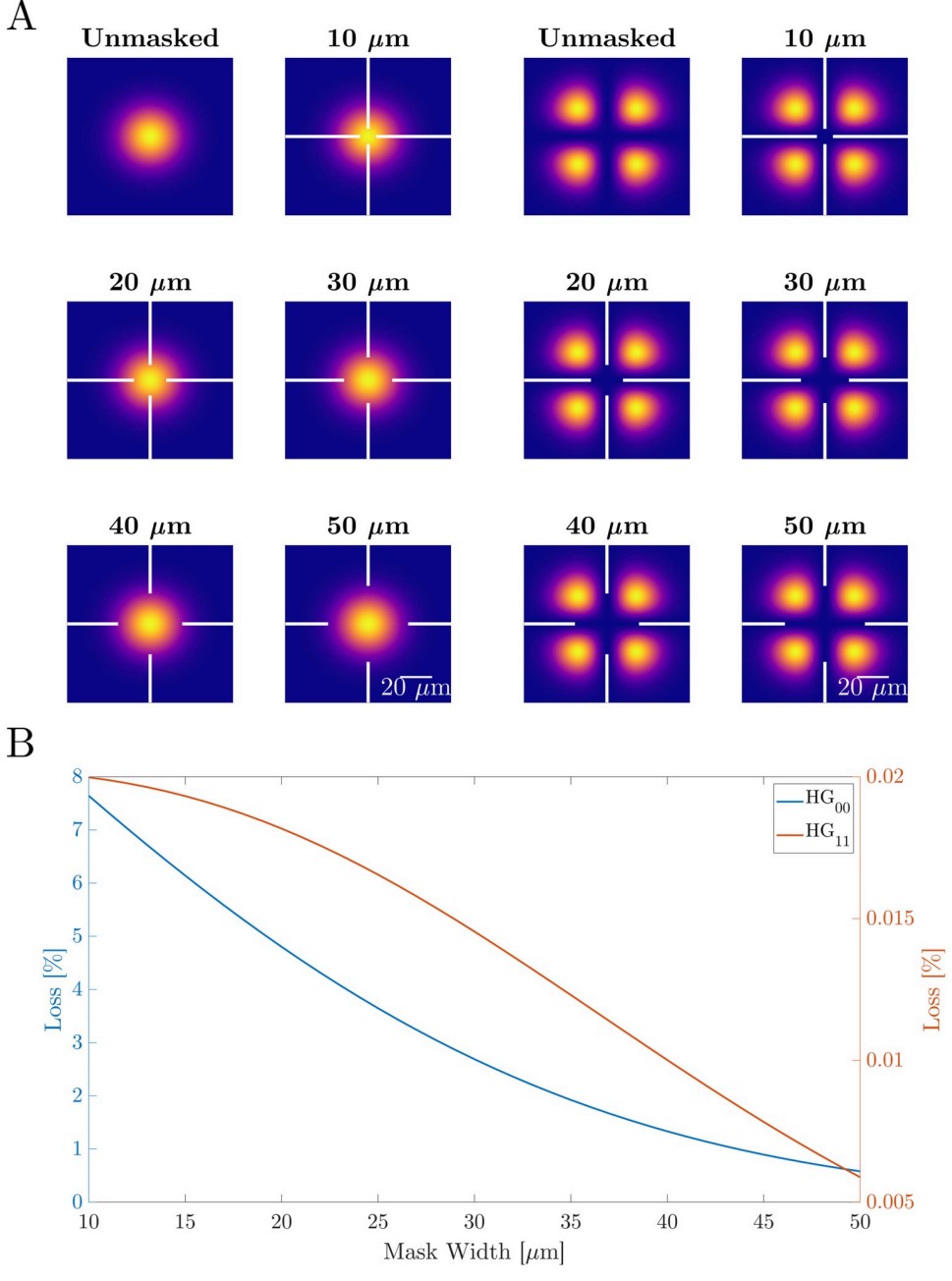

**Fig 3.** A: Simulated overlap between the $HG_{00}$ and $HG_{11}$ modes for each mask width. B: Simulated losses of the $HG_{00}$ and $HG_{11}$ modes as a function of mask width.

spatial measurement of reflectivity of the ablated mirror we set the intensity of masked areas to 6.2% of the calculated value, which represents the reflectivity of the ablated areas and therefore the 93.8% loss on the mode. This way we are able to calculate the expected losses of each mode and can use this as a predictive tool for characterising the bi-frequency operation.

Fig 3B shows the predicted percentage loss of both the $HG_{00}$ and $HG_{11}$ modes for mask widths between 10 and 50 μm. It can be seen that the losses of the $HG_{00}$ mode (blue) are consistently orders of magnitude higher than those of the $HG_{11}$ mode (orange). This is expected as the masks are designed to have minimal overlap with the $HG_{11}$ mode, which has normally less gain due to poorer overlap with the pump beam and higher loss due to higher diffraction. On the 10 μm mask the $HG_{00}$ mode sees 7.64% loss, whereas the $HG_{11}$ mode sees only 0.02%, due to the significantly smaller overlap between the mode and the mask. As the mask size increases the loss of each mode decreases as the overlap between the mask and the areas of higher intensity is minimised. At the 50 μm mask the $HG_{00}$ and $HG_{11}$ modes see 0.58% and 0.006% loss respectively.

### 3.2 MECSEL characterisation

The MECSEL is first characterised while operating on an unablated region of the masked mirror. Fig 4A shows the output power characterisation of the MECSEL as a function of pump power, up to 4.25 W. Fig 4B shows the spectral emission from the MECSEL at 3.34 W of pump power, acquired using a Yokogawa AQ6370D Optical Spectrum Analyzer (OSA). The lasing threshold power for the MECSEL is found to be 1.75 W and a slope efficiency of 8.3% is

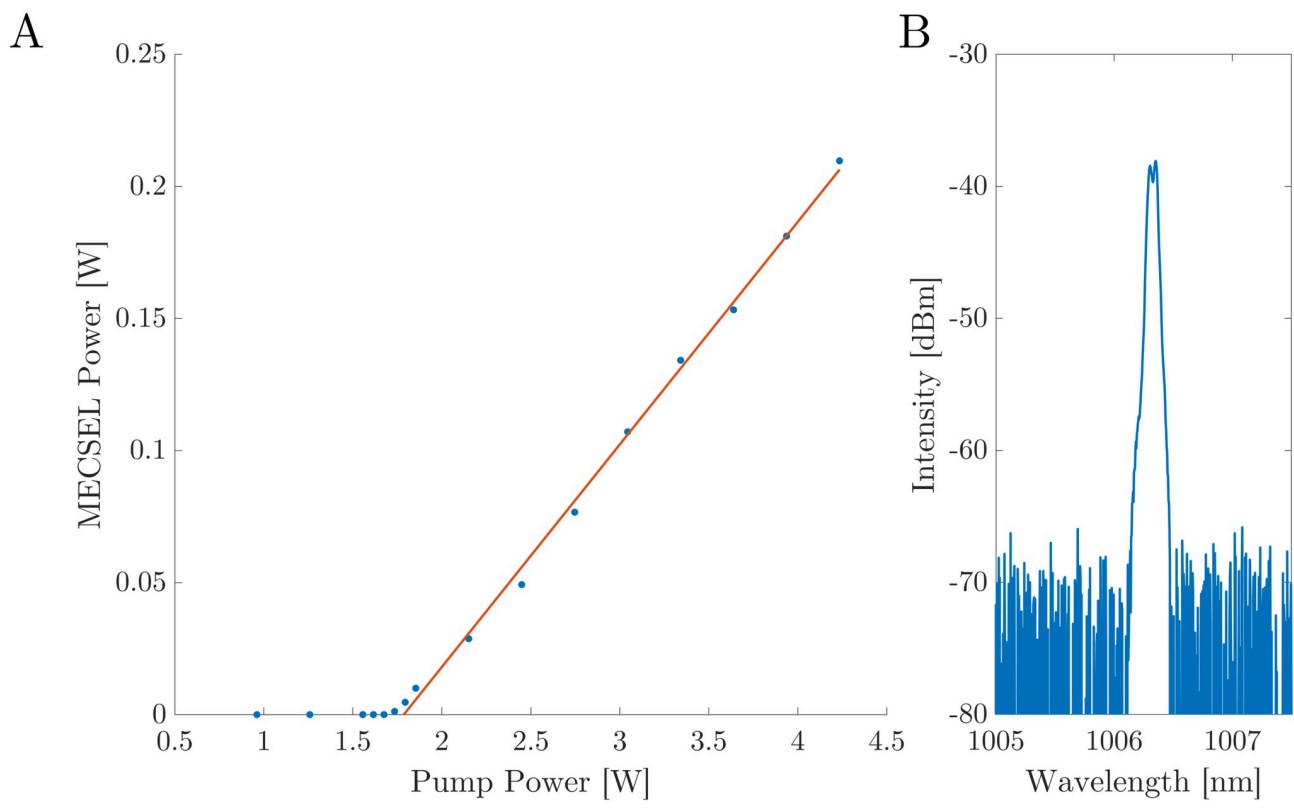

**Fig 4.** A: MECSEL output power calibration as a function of pump power while operating on an unablated region of the masked mirror. The threshold pump power for lasing is seen to be 1.75 W, with a slope efficiency of 8.3%. B: Output spectrum from the MECSEL for a pump power of 3.34 W.

calculated from the fit shown in orange. The MECSEL gain sample used in this work is of the same growth and layer list, as that shown in S. Mirkhanov et al. [24], where an efficiency of 20.3% was shown in a MECSEL cooled to 12˚C. One reason for the deviation in efficiency shown in this work is the large pump spot size necessary for the excitation of the higher order $HG_{11}$ mode, which is $\sqrt{3}$ times larger than the fundamental, therefore causing a large amount of pump energy to be wasted while not exciting the higher order mode. Additionally, by not having the gain sample at the cavity's focus, the intensity on the gain sample is lower which reduces the slope efficiency and increases the threshold.

### 3.3 Bi-frequency operation

Bi-frequency operation is observed in our MECSEL while operating on ablated masks with widths of 20 μm and above. Fig 5 shows the spectral emission from the MECSEL, while operating on each mask and while centred on an unablated region of the masked mirror as a function of pump power, in 60 mW increments, up to a maximum of 3.96 W. While the MECSEL is operating with the cavity spot centred on the 10 μm mask, only the $HG_{11}$ mode is present, meaning that the losses of the $HG_{00}$ mode were sufficient to prevent lasing. The 20 μm mask is the first mask where the cavity is seen to be operating on both modes simultaneously. At lower pump powers, the losses of the $HG_{00}$ mode are still too great to permit lasing, and only the $HG_{11}$ mode is able to operate until a pump power of 3.04 W, after which both modes can be seen to oscillate simultaneously with approximately 5 nm of separation. The spectral tuning of the MECSEL while operating on an unablated region of the masked mirror can be seen to follow a similar trend as the $HG_{00}$ mode. If we use the mode loss estimations in Fig 3B this allows us to estimate that the gain of the $HG_{00}$ mode is between 5.1 and 7.6%, as this is the loss required to prevent lasing.

The threshold pump power of the $HG_{00}$ mode can be seen to decrease as the mask width increases, because the spatially dependent loss of the mode decreases with mask width, as seen in Fig 3B. It can also be seen that the threshold pump power for the $HG_{11}$ mode increases with mask width. We believe that this is due to competition between the two modes for gain where they overlap on the gain sample. On the two largest mask widths, the $HG_{00}$ mode is the first mode to operate in the cavity, due to its greater overlap with the pump beam. The rate of spectral tuning of the $HG_{11}$ modes seen in Fig 5 is predominantly the result of thermal tuning within the pump spot [7, 8]. The separation between the two modes can be seen to remain consistent at approximately 5 nm for the entire range of pump powers on all masks. This is contrary to what we have seen previously in [14] however the cavity and position of the MECSEL gain sample here is quite different.

The wavelength of the MECSEL emission can be seen to increase in set increments rather than smooth tuning; the peak wavelength is found to be increasing in 0.535 ± 0.015 nm increments in both modes as the pump power increases. The resolution of our OSA is 0.02 nm, suggesting that the noise in the wavelength increments is resolution limited but our resolution is accurate enough to resolve the increment reported. A wavelength step of 0.535 ± 0.015 nm corresponds to a frequency difference of 158.8 ± 4.45 GHz, which is approximately equal to the free spectral range (FSR) of the cavity formed in one of the 350 μm thick SiC heatspreaders, and could be explained by the heatspreader acting as an etalon [35, 36]. The finesse of the cavity formed inside the heatspreader is calculated to be approximately 0.85, which we can estimate modulates the transmission of the heatspreader by approximately 20%; this value of wavelength dependent transmission modulation is enough to impact the operation of the laser and force operation at the maxima of the transmission. Fig 6 shows the modulated transmission of the MECSEL due to the etalon in blue, with each of the emitted spectra from the 10 μm

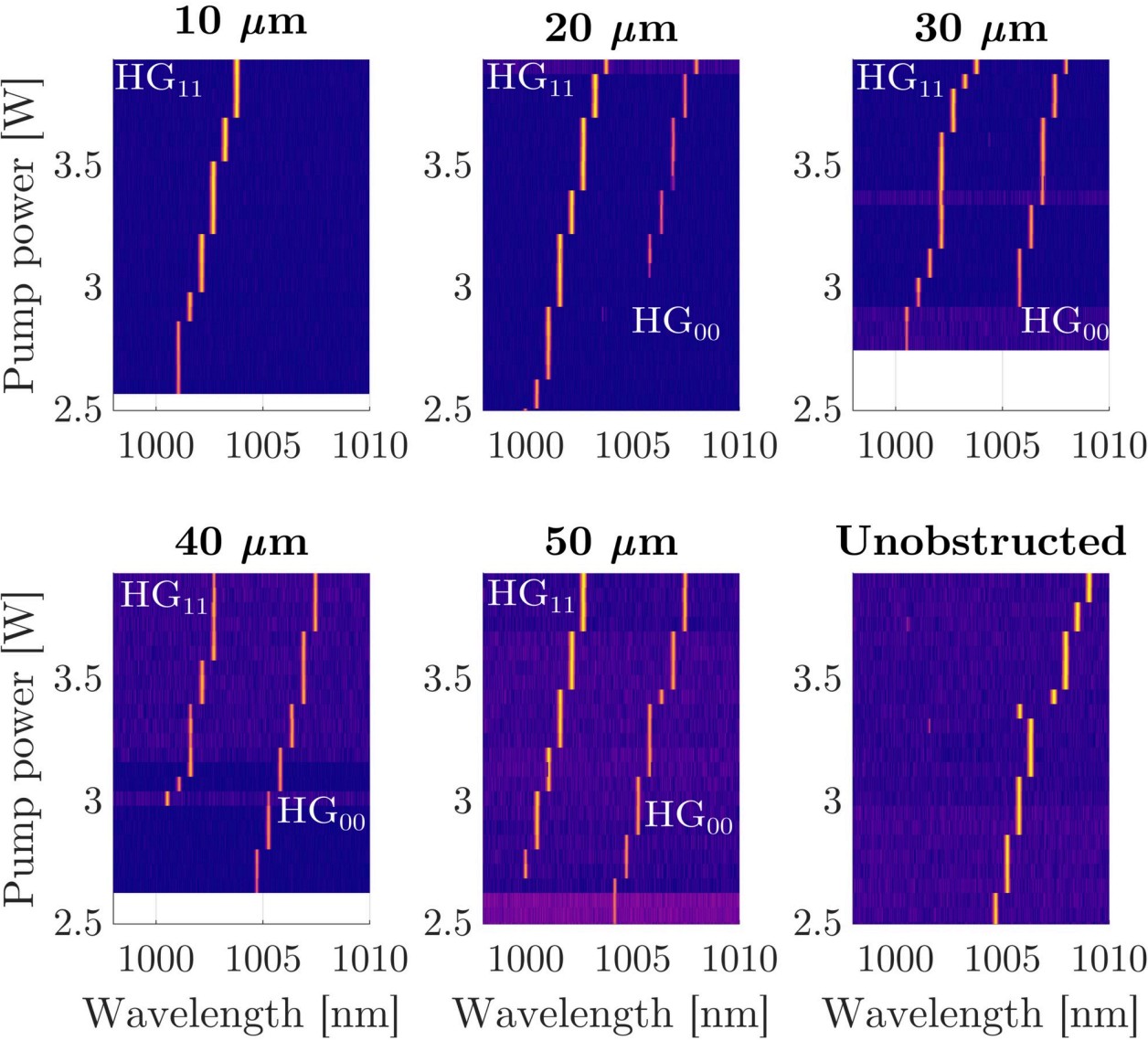

**Fig 5. Spectral emission from the MECSEL as a function of pump power over masks with gap sizes between 10 μm and 50 μm and while operating on an unablated region of the mirror.** The colour map representing the intensity of the spectra is logarithmic, and the noise floor between data sets differs due to the auto-zeroing of the optical spectrum analyser.

mask overlaid in orange, for each of the pump powers that are shown in Fig 5; as expected the MECSEL emission can be seen to only occur around the maxima of the transmission.

Beam profiles are captured using a Hamamatsu ORCA Flash 4.0 digital CMOS camera during MECSEL bi-frequency operation for further characterisation of the laser. Fig 7A shows the experimental beam profile captured while the MECSEL cavity is centred on a 50 μm wide mirror mask, Fig 7B shows the calculated beam profile formed by the combination of the $HG_{00}$ and $HG_{11}$ modes, with the maximum intensity of each mode normalised to 1 in order to display maximum overlap. There is relatively good agreement between the experimental and simulated profiles with only minor aberrations, such as on the bottom left node of the $HG_{11}$ mode, likely due to imperfections in the intra-cavity elements.

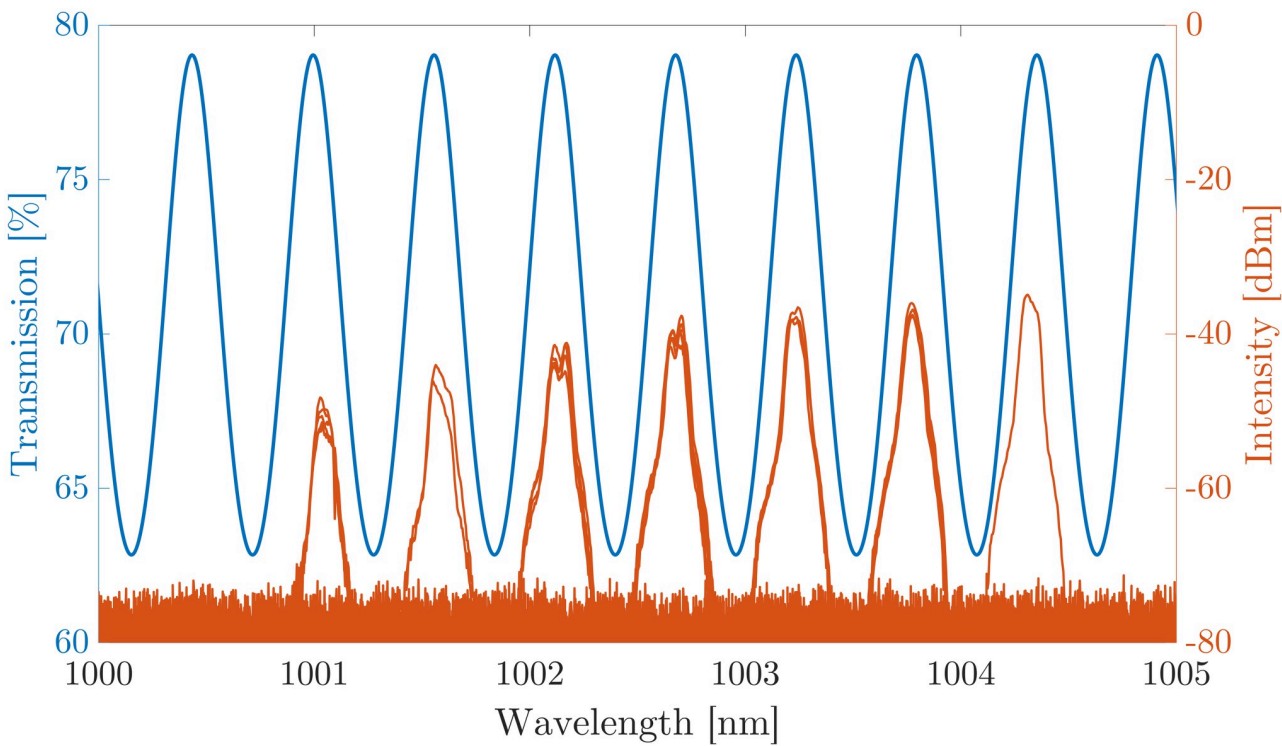

**Fig 6. Heatspreader transmission as a function of wavelength in blue, with the overlaid spectra from the laser operating on the 10 μm mask (orange) for each of the pump powers shown in Fig 5.** It can be seen that lasing only occurs at the maxima of the transmission.

Fig 7C shows the spectral emission from the MECSEL while operating centred on the 50 μm mask at a the maximum pump power of 3.96 W. The wavelength separation between the two modes can be seen to be 4.76 nm, with the $HG_{00}$ centred around 1007.48 nm, and the $HG_{11}$ centred around 1002.72 nm. This separation would correspond to a beat frequency of 1.41 THz, and is relatively consistent over the full range of pump powers and mask gap sizes, as seen in Fig 5.

In order to investigate the spectral purity of the MECSEL while emitting in bi-frequency operation, the laser output is coupled using a multimode fiber to a 25 GHz bandwidth, fiber coupled, fast photodiode, which is connected to an RF spectrum analyser with 40 GHz bandwidth. Fig 8 shows the signal traces of the MECSEL emission while lasing in bi-frequency operation, and while the detector is blocked, as well as the residual of the two traces. There are no features in the RF spectrum, suggesting that there is no beating between adjacent longitudinal modes during operation. In order to study the longitudinal modes operating in the laser, it will be necessary to observe the RF spectrum of the beat between the bi-frequency MECSEL and a stable frequency comb.

## 4 Conclusion

We have demonstrated continuous wave bi-frequency operation in a MECSEL through the use of laser ablated mirror masks on the high reflectivity mirror of the MECSEL external cavity to induce intra-cavity losses. We have shown that the loss induced by the mirror masks, and therefore the lasing thresholds of the effected Hermite-Gaussian modes, are dependent on the size of the unablated central region of our crosshair masks, and thus can be tailored to meet

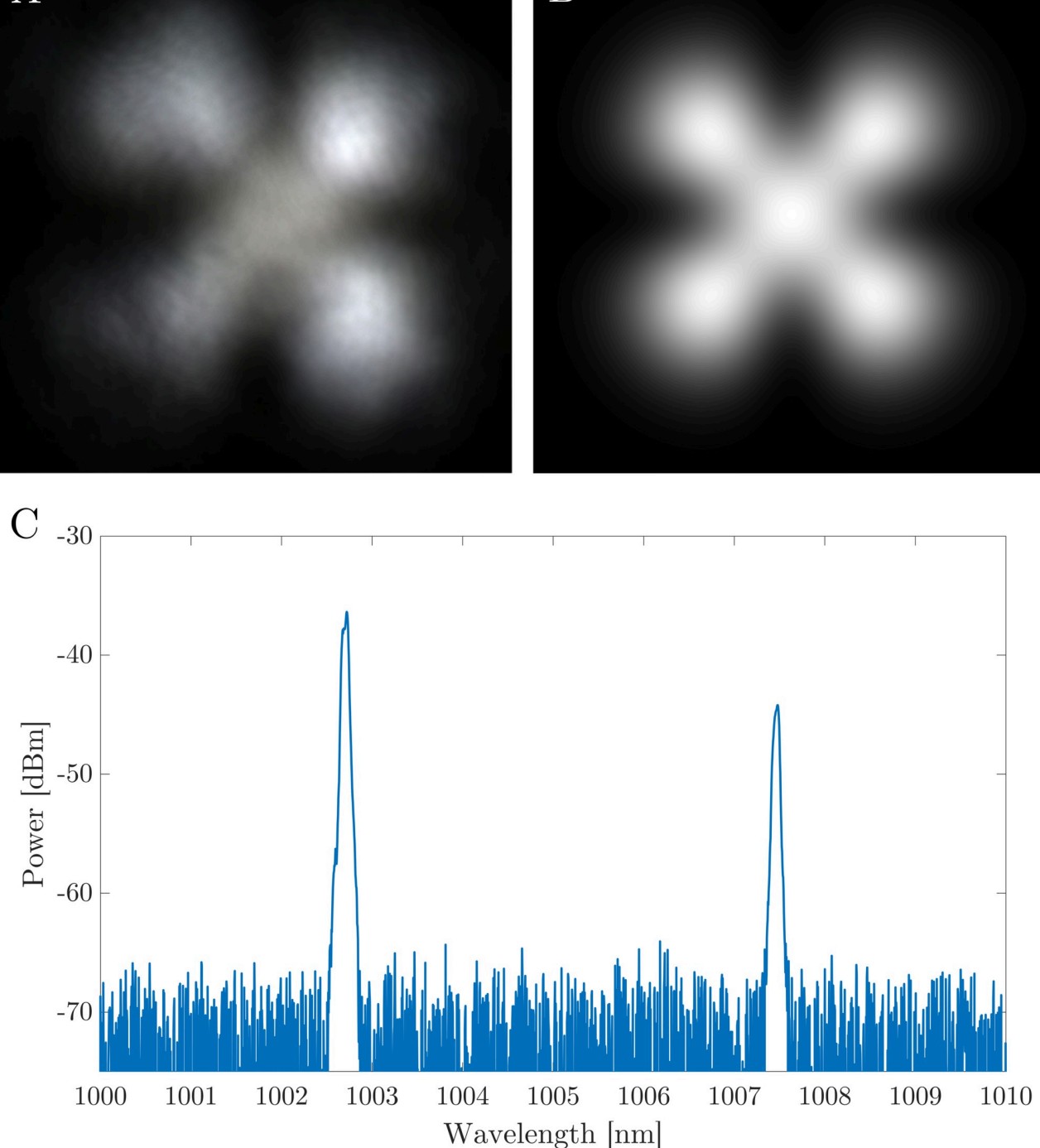

**Fig 7.** A: Beam profile captured using a Hamamatsu ORCA Flash 4.0 CMOS camera during bi-frequency operation of the MECSEL while the cavity is centred on the 50 μm wide ablated mask. B: Calculated combination of $HG_{00}$ and $HG_{11}$ modes. C: Spectral emission from the MECSEL while operating on the 50 μm mask at a pump power of 3.96 W.

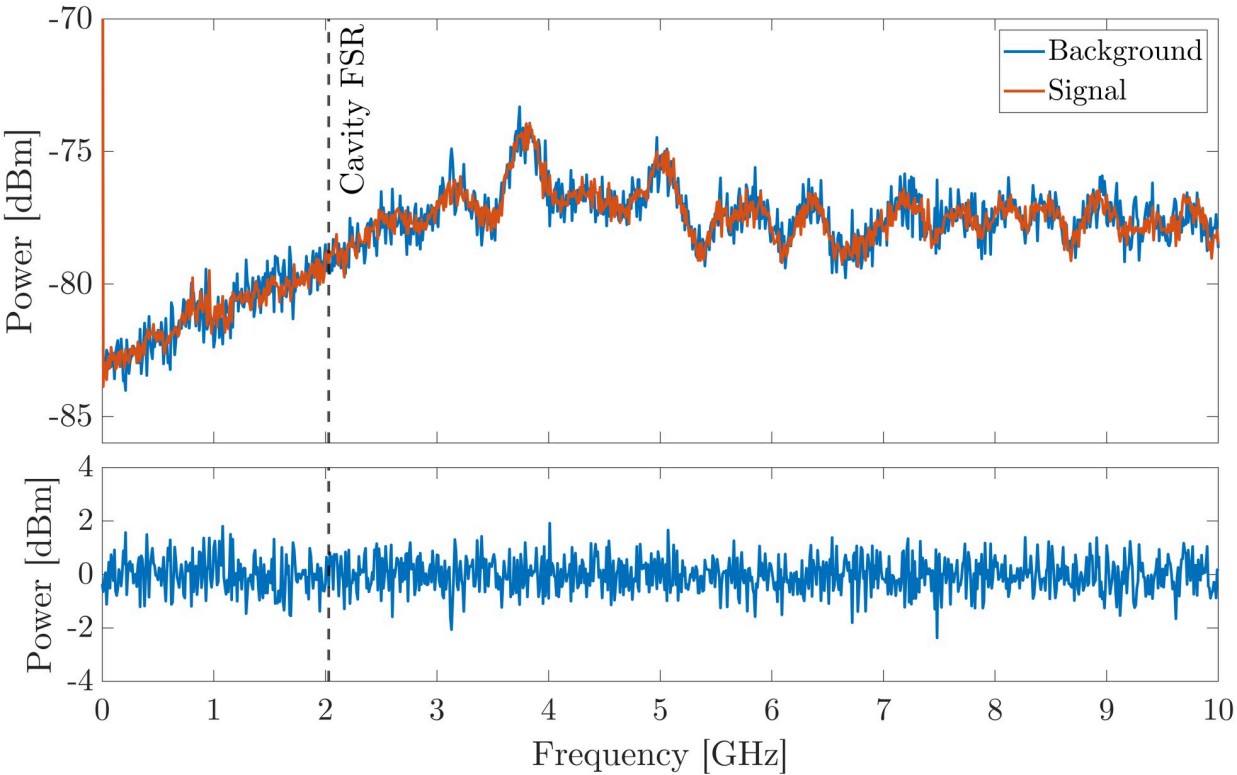

**Fig 8. Top—RF traces taken while the MECSEL is in bi-frequency operation with the detector blocked (blue) and unblocked (orange).** The black dashed line represents the cavity free spectral range of 2.03 GHz. Bottom—Residual of the signal and background traces.

specific criteria. The fact that the ablated masks are separate from the gain sample, gives us the flexibility for the same laser cavity to be used with different sets of mirror masks in order to investigate different modes of operation. In future the loss inducing masks could be relocated from a cavity mirror onto one of the gain sample heat spreaders to reduce complexity in the cavity and allow for a more conventional laser cavity where we could place the MECSEL gain sample in the focus for much more efficient and powerful operation. MECSELs have shown that they can be operated in a large bandwidth of nearly 25 THz [27]. In future we would want to use this advantage and achieve tuning of the separation of the emission frequencies to use a similar system in order to generate THz spectrum using photomixers that can be narrow line-width and spanning a wide range of the THz spectrum.

## Acknowledgments

The authors would like to thank Daniel Heath for his work in fabricating the masked mirror.

## Author Contributions

**Conceptualization:** Jake Daykin, Jonathan R. C. Woods, Vasilis Apostolopoulos.

**Data curation:** Jake Daykin.

**Formal analysis:** Jake Daykin.

**Funding acquisition:** Vasilis Apostolopoulos.

**Investigation:** Jake Daykin, Jonathan R. C. Woods.

**Methodology:** Jake Daykin, Jonathan R. C. Woods.

**Resources:** Roman Bek, Michael Jetter, Peter Michler, Ben Mills.

**Supervision:** Jonathan R. C. Woods, Peter Horak, James S. Wilkinson, Vasilis Apostolopoulos.

**Writing – original draft:** Jake Daykin.

**Writing – review & editing:** Jake Daykin, Jonathan R. C. Woods, Roman Bek, Ben Mills, Peter Horak, James S. Wilkinson, Vasilis Apostolopoulos.

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
