## [Decision Letter · Decision Letter 0]

18 Jun 2023

PONE-D-23-16239Bi-frequency operation in a membrane external-cavity surface-emitting laserPLOS ONE

Dear Dr. Daykin,

Thank you for submitting your manuscript to PLOS ONE. After careful consideration, we feel that it has merit but does not fully meet PLOS ONE’s publication criteria as it currently stands. Therefore, we invite you to submit a revised version of the manuscript that addresses the points raised during the review process.

 Please submit your revised manuscript by Aug 02 2023 11:59PM. If you will need more time than this to complete your revisions, please reply to this message or contact the journal office at plosone@plos.org. Please include the following items when submitting your revised manuscript:A rebuttal letter that responds to each point raised by the academic editor and reviewer(s). You should upload this letter as a separate file labeled 'Response to Reviewers'.A marked-up copy of your manuscript that highlights changes made to the original version. You should upload this as a separate file labeled 'Revised Manuscript with Track Changes'.An unmarked version of your revised paper without tracked changes. You should upload this as a separate file labeled 'Manuscript'.If applicable, we recommend that you deposit your laboratory protocols in protocols.io to enhance the reproducibility of your results. Protocols.io assigns your protocol its own identifier (DOI) so that it can be cited independently in the future. For instructions see: https://journals.plos.org/plosone/s/submission-guidelines#loc-laboratory-protocols. Additionally, PLOS ONE offers an option for publishing peer-reviewed Lab Protocol articles, which describe protocols hosted on protocols.io. Read more information on sharing protocols at https://plos.org/protocols?utm_medium=editorial-email&utm_source=authorletters&utm_campaign=protocols.

We look forward to receiving your revised manuscript.

Kind regards,

Xuejian Wu, Ph.D.

Academic Editor

PLOS ONE

Journal Requirements:

Reviewers' comments:

Reviewer's Responses to Questions

**Comments to the Author**

1. Is the manuscript technically sound, and do the data support the conclusions?

Reviewer #1: Yes

Reviewer #2: Yes

Reviewer #3: Yes

2. Has the statistical analysis been performed appropriately and rigorously? 

Reviewer #1: Yes

Reviewer #2: Yes

Reviewer #3: Yes

3. Have the authors made all data underlying the findings in their manuscript fully available?

Reviewer #1: Yes

Reviewer #2: Yes

Reviewer #3: Yes

4. Is the manuscript presented in an intelligible fashion and written in standard English?

Reviewer #1: Yes

Reviewer #2: Yes

Reviewer #3: Yes

5. Review Comments to the Author

Reviewer #1: Please see the attached file.

Reviewer #2: In this article, the authors report continuous wave bi-frequency operation in a membrane external-cavity surface-emitting laser (MECSEL). By controlling spatially specific loss on certain transverse cavity modes, the authors successfully operated the laser on two Hermite-Gaussian spatial modes simultaneously. The experimental results support their findings, showcasing bi-frequency operation across different parameters. The clear description and substantial results demonstrate the quality and significance of the research. I would like to recommend the manuscript to be accepted for publication. I have a couple of minor comments:

1.) Could the author clarify more about how does this work differs from previous work? Other than using a ablation pattern, are there any other methods for bi-frequency operation? It would be more informative to brief talk about this in the introduction.

2.) What’s the beam radii of even higher order modes (other than HG00 and HG11)? It would be helpful to add this information in section 2.3.

Reviewer #3: The paper titled "Bi-frequency operation in a membrane external-cavity surface-emitting laser " by Jake Daykin et al demonstrates the continuous wave bi-frequency operation in a membrane external-cavity surface-emitting laser. By introducing spatially specific loss to the intra-cavity high reflectivity mirror with laser ablated mirror masks, the laser simultaneously operates on two Hermite-Gaussian spatial modes. The bi-frequency operation is demonstrated across different pump powers and sizes of spatial loss features, with a wavelength separation of approximately 5nm centered at 1005 nm.

The paper is properly structured and well written. The experiments in section 2 and 3 are well designed and convincing. The idea of utilizing spatially dependent loss within the MECSEL cavity to control transverse cavity modes and facilitate the oscillation of two specific spatial modes is new to the field. Therefore the paper in my opinion certainly should be considered for publication in PLOS ONE.

There is one minor issue to address: On page 1, line 7, please write out the full expressions of VECSEL on first use.

6. PLOS authors have the option to publish the peer review history of their article (what does this mean?). If published, this will include your full peer review and any attached files.

Reviewer #1: No

Reviewer #2: No

Reviewer #3: **Yes: **Weicheng Zhong

---

## [Author Response · Author response to Decision Letter 0]

10 Jul 2023

Dear Editor,

Thank you for your latest correspondence regarding the peer review decision for manuscript PONE-D-23-16239, titled “Bi-frequency operation in a membrane external-cavity surface-emitting laser”

First of all, I would like to thank the reviewers for their time in reviewing our manuscript which helped us to improve the quality of the manuscript. In response to the feedback from the reviewers, we have taken the time to improve the manuscript and would like to take the opportunity to respond to each of the comments presented by the reviewers.

Reviewer 1: 

 In this paper, the authors demonstrated an optically pumped bi-frequency MECSEL. One of the cavity mirrors is laser ablated to introduce a transverse-mode-dependent loss, making it possible for two transverse modes to lase simultaneously with similar output power. 

The manuscript introduces details on the laser structure including the ablated mirror, cavity, gain medium, and pump laser. Operations under various pump power and different ablated masks are experimentally studied. The results show promising bi-frequency operations with tuning capabilities. I would like to recommend the manuscript be published after addressing a few questions below: 

1. The acronym “VECSEL” first appears in line 7, while the full term “Vertical External-Cavity Surface-Emitting Laser” appears later in line 14. Please make sure the acronym is defined at the first appearance. 

2. In line 61, the masks are described as “with ablated central regions”. However, the actual masks seem to have unablated central regions. Can the authors clarify this? 

3. In line 132, the authors mentioned Fig. 5 has a 300-mW increment of pump power. The pump power range in Fig. 5 plots is about 2.5 ~ 4.0 W, so there should be only 5 or 6 discrete pump power values. However, some plots in Fig. 5 show more than 6 different spectrum sections across the pump power range, so it seems the increment is less than 300 mW. Can the authors clarify this? 

4. In Fig. 5, is the colormap in log scale or linear scale? Some plots seem to have a higher noise floor at some pump power values (for example, the 50 um mask plot at about 2.5 W pump power). What is the reason for this? 

We thank the reviewer for their thorough comments and have addressed them in the following ways: 

 Vertical External-Cavity Surface-Emitting Lasers defined on line 7 instead of line 14/15.

 Corrected typo of “ablated” to “unablated” on line 68.

 Corrected typo on line 150. Power increments are 60mW, not 300mW.

 The caption for Figure 5 has been altered to include the following: 

“The colour map representing the intensity of the spectra is logarithmic, and the noise floor between data sets differs due to the auto-zeroing of the optical spectrum analyser.”

Reviewer 2: 

In this article, the authors report continuous wave bi-frequency operation in a membrane external-cavity surface-emitting laser (MECSEL). By controlling spatially specific loss on certain transverse cavity modes, the authors successfully operated the laser on two Hermite-Gaussian spatial modes simultaneously. The experimental results support their findings, showcasing bi-frequency operation across different parameters. The clear description and substantial results demonstrate the quality and significance of the research. I would like to recommend the manuscript to be accepted for publication. I have a couple of minor comments:

1.) Could the author clarify more about how does this work differs from previous work? Other than using a ablation pattern, are there any other methods for bi-frequency operation? It would be more informative to brief talk about this in the introduction. 

2.) What’s the beam radii of even higher order modes (other than HG00 and HG11)? It would be helpful to add this information in section 2.3.

We would like to thank the reviewer, and provide the following responses to their comments:

 A summary of the mechanisms used for creating a bi-frequency output is described in the second paragraph of the introduction, they include ablation of cavity mirrors, metalisation of the gain sample, birefringent filters in the cavity. The work described in this manuscript differs from previous work as we have implemented bi-frequency operation on the MECSEL Laser platform. The bi-frequency MECSEL presented is, to our knowledge, the first demonstration of a MECSEL operating on two frequencies simultaneously. Moving this technology to a MECSEL from the traditionally used VECSEL gives many opportunities as explained in the introduction and conclusions. MECSELs because of the removal of the intracavity DBR have the following advantages: faster to grow, better thermal management, greater wavelength selectability. 

We have altered the introduction of the paper to better highlight the previous work of generating bi-frequency operation in the VECSEL platform and explain the difference in our work.

 The radius (Rn,m) of a higher order mode is given by: R_(n,m)=R_00 √(1+n+m).Where R00 is the radius of the fundamental mode. 

We agree with the reviewer’s comment and have altered Section 2.3 to include the above relation and we have included a clarification that the HG 00, 11, 33, 55 etc modes would be able to operate within the mask design. However, modes higher order than the 11 will be suppressed by diffraction losses and by choice of cavity parameters. 

Reviewer 3: 

The paper titled "Bi-frequency operation in a membrane external-cavity surface-emitting laser " by Jake Daykin et al demonstrates the continuous wave bi-frequency operation in a membrane external-cavity surface-emitting laser. By introducing spatially specific loss to the intra-cavity high reflectivity mirror with laser ablated mirror masks, the laser simultaneously operates on two Hermite-Gaussian spatial modes. The bi-frequency operation is demonstrated across different pump powers and sizes of spatial loss features, with a wavelength separation of approximately 5nm centered at 1005 nm.

The paper is properly structured and well written. The experiments in section 2 and 3 are well designed and convincing. The idea of utilizing spatially dependent loss within the MECSEL cavity to control transverse cavity modes and facilitate the oscillation of two specific spatial modes is new to the field. Therefore the paper in my opinion certainly should be considered for publication in PLOS ONE.

There is one minor issue to address: On page 1, line 7, please write out the full expressions of VECSEL on first use.

We thank the reviewer for their time and recommendation, and have addressed their raised issue:

 Vertical External-Cavity Surface-Emitting Lasers defined on line 7 instead of line 14/15

We thank the reviewers that helped us to improve the quality of the manuscript and I hope that our responses to the reviewers’ comments have been sufficient to justify the publication of our manuscript in PLOS ONE.

Sincerely,

Jake Daykin

School of Physics and Astronomy

University of Southampton

---

## [Editor Report · Decision Letter 1]

14 Jul 2023

Bi-frequency operation in a membrane external-cavity surface-emitting laser

PONE-D-23-16239R1

Dear Dr. Daykin,

We’re pleased to inform you that your manuscript has been judged scientifically suitable for publication and will be formally accepted for publication once it meets all outstanding technical requirements.

Kind regards,

Xuejian Wu, Ph.D.

Academic Editor

PLOS ONE
---

## [Editor Report · Acceptance letter]

19 Jul 2023

PONE-D-23-16239R1 

Bi-frequency operation in a membrane external-cavity surface-emitting laser 

Dear Dr. Daykin:

I'm pleased to inform you that your manuscript has been deemed suitable for publication in PLOS ONE. Congratulations! Your manuscript is now with our production department. 

Kind regards, 

on behalf of

Dr. Xuejian Wu 

Academic Editor

PLOS ONE